# Disordered Eating Behaviors and Weight Regain in Post-Bariatric Patients

**DOI:** 10.3390/nu16234250

**Published:** 2024-12-09

**Authors:** Karynne Grutter Lopes, Eline Coan Romagna, Diogo Menezes Ferrazani Mattos, Luiz Guilherme Kraemer-Aguiar

**Affiliations:** 1Postgraduate Program in Clinical and Experimental Physiopathology, Faculty of Medical Sciences, State University of Rio de Janeiro, Rio de Janeiro 20550-013, RJ, Brazil; kjgolrj@gmail.com (K.G.L.); elinecr@gmail.com (E.C.R.); 2Obesity Unit, Multiuser Clinical Research Center (CePeM), Pedro Ernesto University Hospital, State University of Rio de Janeiro, Rio de Janeiro 20550-013, RJ, Brazil; 3MídiaCom/Postgraduate Program on Electrical and Telecommunications Engineering, Fluminense Federal University, Niteroi 24020-140, RJ, Brazil; 4Endocrinology, Department of Internal Medicine, Faculty of Medical Sciences, State University of Rio de Janeiro, Rio de Janeiro 20550-013, RJ, Brazil

**Keywords:** bariatric surgery, eating disorder, weight regain, time since surgery

## Abstract

Background/objective: Weight regain has serious health consequences after bariatric surgery, and disordered eating behaviors (EBs) may be involved in it. We compared disordered EB symptoms in post-bariatric patients according to low vs. high ratio of weight regain (RWR) and investigated associations between disordered EB symptoms with weight regain and time since surgery. Method: We recruited ninety-four patients who had undergone laparoscopic *Roux-en-Y* gastric bypass or sleeve gastrectomy. All of them had not attended follow-up with the multidisciplinary healthcare team (including psychological assessment) for at least one year. RWR was calculated with respect to maximal weight loss by the nadir weight achieved after surgery. Patients were divided into two groups: high RWR (≥20%) or low RWR (<20%). At their first visit, we had them complete the Eating Disorder Examination and Repetitive Eating Questionnaires (EDE-Q and Rep(eat)-Q). Results: Patients with high RWR reported higher EDE-Q global, dietary restraint and weight concerns without significant group differences in eating and shape concerns. Compared to those patients with low RWR, these patients also indicated greater Rep(eat)-Q global, compulsive grazing and repetitive eating. Global EDE-Q score, eating concern, shape concern, weight concern and all Rep(eat)-Q indexes were positively associated with higher RWR. Conclusions: Disordered EBs occurred more frequently in the group with high RWR. In these patients, eating behavior symptoms and grazing behavior were both correlated to the RWR, suggesting a possible involvement of both conditions in weight regain.

## 1. Introduction

Obesity is increasing its prevalence over the years and is an important public health problem worldwide [1]. Bariatric surgery is an effective and sustained treatment for severe obesity [2], improving or remitting chronic diseases and reducing mortality [3]. Even after achieving significant weight loss after surgery, 10–20% of these patients may regain weight over the years [4]. Regaining weight after surgery has serious health consequences. It may lead to obesity-related comorbidity recidivism, such as type 2 diabetes *mellitus* (T2D) [5], psychological distress, poorer health-related quality of life and also an economic burden due to the need for new treatments, including antiobesity drugs and even the need for revisional surgeries [6].

Himpens et al. emphasized one crucial aspect: weight regain coincided with discharge from regular follow-ups with the multidisciplinary healthcare team (MHT) three years after surgery [7]. It is well-recognized that MHT care for post-bariatric patients may improve outcomes [8]. However, in the long-term, the management of these patients is less common, and they may be less likely to comply with the MHT’s recommendations.

Some causes of weight regain are relevant, such as hormonal or metabolic changes, anatomical factors, lifestyle changes, low levels of physical activity, non-adherence to diet and eating disorder (ED) symptoms [9]. After surgery, some patients may develop disordered eating behaviors (EBs), or the pre-existing ones may return, such as loss of control, overeating, binge eating, night eating and grazing [10]. These disordered EBs can lead to poor weight outcomes [11], decreasing mental health-related quality of life [12]. The presence of ED symptoms and their influences on weight regain is not fully elucidated, especially in those patients without MHT follow-up post-surgery [13].

Emerging evidence suggests that greater attention is warranted since disordered EBs might hinder post-surgical weight loss in the short and long term (six months and 2 to 6 years, respectively) [14,15,16]. Overvaluation of weight/shape appears to be associated with more significant ED psychopathology [17]. The findings of Conceição et al. [18] corroborated these assertions, demonstrating that disordered EBs predicted greater weight regain two years subsequent to gastric bypass and gastric banding procedures.

The increasing number of procedures performed nowadays fosters the need to study these patients, their eating behaviors and clinical aspects, and the impact on post-surgical outcomes for better monitoring by assisting MHTs. The treatment strategies for weight regain must be updated and evaluated according to guidelines and the needs of each patient. Thus, we compared ED symptoms (disordered EBs and grazing) in patients who had undergone a bariatric surgery and did not attend MHT follow-up. These patients were compared according to low vs. high ratio of weight regain (RWR). We also tested the associations between ED symptoms vs. weight regain and time since surgery.

## 2. Materials and Methods

### 2.1. Patients and Data Assessment

Of the 132 recruited, 94 patients (87.2% females, aged 42 ± 9 years and body mass index (BMI) 32.9 ± 6.5 kg/m^2^) were included. Of these, 80 underwent Roux-en-Y Gastric Bypass (RYGB) and 14 had a Sleeve Gastrectomy (SG). Twenty-one declined to participate, one was pregnant, six lost follow-ups and 10 had less than 24 months of surgical procedure. The time since their surgery was 6.1 ± 4.0 years, and almost 8 in 10 (79.8%) patients had their surgeries in the private healthcare system but lost follow-up. Patients were summoned through social media to restart the standard clinical follow-up with our MHT (composed of surgeons, endoscopists, endocrinologists, dieticians, psychologists, psychiatrists, physiotherapists and physical educators) with expertise in bariatric patients, following the clinical practice guidelines [8]. On their first visit to our unit, they were all invited to be part of this study. As previously described, they entered our public healthcare unit for regular follow-up after this first visit [19]. Only patients who underwent laparoscopic RYGB or SG, achieving a mean 50% weight loss, and had a follow-up gap of at least one year with the MHT were included. Pregnancy, physical activity level of ≥150 min/week, use of any medication affecting weight, not having autonomy for diet choices, history of revisional surgery, having less than 24 months from surgery and missing data on pre- and post-operative anthropometry, as well as on surgical procedure details, inability to understand/read Portuguese texts and unanswered or incomplete questionnaires were exclusion criteria employed. A digestive endoscopy was requested for all patients to ensure the informed surgical type.

We recruited and screened participants between November 2018 and December 2019. Clinical history, physical exam, anthropometrics, blood pressure and heart rate were collected as previously described [19]. Glycated hemoglobin (A1C) ≥ 6.5% or fasting plasma glucose ≥ 126 mg/dL [20], blood pressure ≥ 130 or 80 mmHg [21] and low-density lipoprotein-cholesterol ≥ 160 mg/dL, or triglyceride levels ≥ 150 mg/dL, or high-density lipoprotein-cholesterol < 40 or <50 mg/dL (for men or women, respectively) [22] were used to categorize the presence of T2D, hypertension and dyslipidemia, respectively. Afterward, patients completed the self-report measures to assess symptoms suggestive of an ED and the presence of grazing.

### 2.2. Data on Bariatric Outcomes

Preoperative and minimum postoperative weights were self-reported and used to calculate BMI. These self-reports were verified via a follow-up call six months post-initial visit. Excess weight loss (EWL) and weight regain (RWR) were calculated as follows: (a) EWL = [(preoperative weight − nadir weight)/(preoperative weight − ideal weight for BMI 25)] × 100%; (b) RWR = [(current weight − nadir weight)/(preoperative weight − nadir weight)] × 100%. Even knowing that there are many definitions of suboptimal weight outcomes post-surgery, we opted to categorize patients according to the weight regained [23]. High RWR was defined as a ≥20% increase from the nadir weight achieved 12–18 months post-surgery [24]. Nadir weight was defined as the lowest weight the participant achieved after surgery. Patients were allocated into low (<20%) or high (≥20%) RWR groups.

### 2.3. Anthropometry and Blood Pressure Assessments

Body mass and height were measured using a calibrated electronic scale and stadiometer (Welmy™ W300A, São Paulo, SP, Brazil), and BMI was calculated. Neck circumference was measured at the inferior margin of the laryngeal prominence, perpendicular to its long axis. Waist circumference was measured at the umbilicus at end-expiration, while hip circumference at the widest point around the gluteal region. Blood pressure and heart rate were measured using a standardized sphygmomanometer (G-Tech^TM^ BSP11, Hangzhou, Zhejiang, China), according to American Heart Association guidelines [25].

### 2.4. Eating Disorder Examination Questionnaire (EDE-Q)

The Portuguese version of the Eating Disorder Examination Questionnaire (EDE-Q) was employed to evaluate the suggestive and associated psychological characteristics of eating disorder symptoms. This instrument enabled the calculation of four subscale scores: dietary restraint, eating concern, weight concern and shape (self-image) concern [26,27]. Dietary restraint involves intentional food restriction for shape and weight reasons. Concurrently, the other three subscales reflect maladaptive eating attitudes and an overemphasis on weight and shape. Twenty-eight EDE-Q items were rated using 0–6 points. Higher scores correspond to a greater severity or frequency of ED psychopathology. The EDE-Q evaluates three distinct types of binge eating: objective bulimic episodes, characterized by the consumption of a substantial amount of food accompanied by a sense of loss of control; subjective bulimic episodes, characterized by the consumption of a relatively small amount of food accompanied by a sense of loss of control; and objective overeating, defined by the consumption of a substantial quantity of food without a sense of diminished control. EDE-Q also includes questions about self-induced vomiting and intensive exercise as methods for shape or weight control [26]. Gero et al. [28] cut-off point was applied, considering total score values < 2.5 to define the absence of symptoms for an ED, classifying them into healthy and unhealthy subgroups.

### 2.5. Repetitive Eating Questionnaire (Rep(eat)-Q)

The Rep(eat)-Q was employed to investigate and screen grazing habits, characterized by unplanned, repetitive consumption of small amounts of food and/or eating that is not responsive to hunger/satiety sensations. The frequency of these behaviors over the preceding month was assessed using a Likert scale ranging from 0 (never) to 6 (every day), resulting in a global score and two subscales: compulsive grazing, defined by a loss of control over eating and its repetitive, distracted nature; and non-compulsive grazing, characterized by repetitive eating without a sense of loss of control [29].

### 2.6. Ethical Approval

This cross-sectional study was approved by the local Ethics Committee (CAAE: 07662918.1.0000.5259) and registered on ClinicalTrials.gov (NCT04193384, accessed on 22 March 2022). All procedures were performed according to the principles of the Declaration of Helsinki. A signed consent form was obtained from each participant.

### 2.7. Statistical Analysis

Data normality was tested using the Shapiro–Wilk test. Differences between groups were analyzed using either the unpaired Student *t*-test or the Mann–Whitney U test, as appropriate. The Chi-square test was used to examine the differences between the distributions of categorical variables. Results were expressed as means ± standard deviations, medians (interquartile ranges) or frequency (counts and proportions). Bonferroni correction was used for multiple comparisons. Pearson correlation coefficients were calculated to investigate the associations between the self-reported measures and ED symptoms vs. weight regain and time since surgery. All calculations were analyzed using NCSS^TM^ statistical software (Version 1, LLC, Kaysville, UT, USA). The significance level was set at *p* ≤ 0.05.

## 3. Results

Clinical and surgery data of patients are exhibited in Table 1. The groups were similar, except for BMI, body circumferences, diastolic blood pressure (DBP), RWR and time since surgery, which were higher in those with high vs. low RWR (*p ≤* 0.05). A significantly higher percentage of patients with dyslipidemia was detected in the high-RWR group (*p* = 0.01). Higher associations between the two self-reported preoperative and nadir weights (r = 0.97 and r = 0.94; *p* = 0.001) validated patient self-reported data.

Table 2 exhibits ED symptoms and the screening of grazing behavior. Compared to patients with low RWR, those with high RWR exhibited higher scores in the global EDE-Q (0.18 to 5.75 points), dietary restraint (0 to 18 points) and weight concern (0 to 6 points; *p ≤* 0.02), as well as in global Rep(eat)-Q score, compulsive grazing and repetitive eating subscales (0 to 6 points each; *p ≤* 0.03). Eating and shape concerns were not different between groups, with a range of 0.1 to 6 points (*p ≥* 0.08). In addition, comparable numbers and percentages of patients reported experiencing objective and subjective bulimic episodes, objective overeating, self-induced vomiting and intense exercise at least once within the preceding month, regardless of their level of weight regain (low vs. high RWR, *p ≥* 0.08).

To investigate these patients further, we correlated their weight regain and disordered EBs and possibly the effects of the time since surgery on these outcomes. Table 3 presents these correlations stratified into pooled samples and according to RWR groups. In the pooled sample, global EDE-Q score, eating concern, shape concern and weight concern were positively correlated to RWR, and dietary restraint was positively correlated to time since surgery (*p ≤* 0.03). Global Rep(eat)-Q score and their two subscales (compulsive grazing and repetitive eating) were positively correlated to RWR and time since surgery (*p ≤* 0.03). In the low-RWR group, the global Rep(eat)-Q score and their two subscales were positively correlated only with RWR (*p ≤* 0.01). Regarding the high-RWR group, the global EDE-Q score, eating concern, shape concern, weight concern and all Rep(eat)-Q indexes were positively correlated with RWR (*p ≤* 0.05). In addition, a direct correlation was present for the repetitive eating subscale with the time since surgery (*p* = 0.04).

## 4. Discussion

The present study enrolled patients who had undergone bariatric surgery and had not attended follow-up consultations with the MHT for a minimum of one year. It also compared those with low vs. high RWR to investigate the role of disordered EBs on RWR. Eating disorder symptoms were observed more frequently in patients who exhibited greater weight regain. Significant clinical differences between groups were also evident in terms of BMI, body circumferences, diastolic BP and dyslipidemia, possibly reflecting the impact of regaining weight after effective bariatric surgery [4]. The self-reported EWL, greater than 50% of our patients, may confirm the latter assertion [30].

The EDE-Q showed significant differences in the global score, which was higher in the high vs. low RWR. Our patients had higher global scores of symptoms suggestive of an ED than healthy young individuals [31] and patients after bariatric surgery in their first year postoperative [28]. Among the candidates for bariatric surgery, we usually diagnose and treat those with ED symptoms. Disordered EB is relatively common in pre-surgical populations [10,32], but there is still limited evidence for the value of diagnosing and treating it before surgery and for the prognostic weight outcomes. Some authors have shown that patients who exhibited slower weight loss trajectories in the first postoperative year had more preoperative psychopathology [33]. In contrast, others [18] observed that patients with preoperative disordered EBs did not differ in weight loss trajectories across 30 months post-surgery. One aspect observed in clinical practice and already studied is that the disordered EB tends to reduce in the first year after surgery [28]. However, it relapses due to a lack of nutritional and psychological counseling and follow-up for more healthy food choices, understanding physical/emotional hunger and satiety, controlled food intake [34] and even the need for drugs to treat them. Although our patients had a mean of 6.1 years of surgery, they were without MHT follow-up for at least one year before enrollment in the present study. It is unknown if/what level of care was provided in the interim between bariatric surgery and their first visit in this study, and we can only speculate that ED symptoms may be one of the factors that contributed to weight regain. One point that ensures our speculation mentioned above is that despite the absence of preoperative data on EB, both groups had the same weight trajectories expressed by similar EWL.

Higher EDE-Q subscale values and percentage of patients with eating and compensatory behaviors have been observed in our sample. Significant changes between the high- and low-RWR groups were dietary restraint and weight concern. Although no difference between study groups has been found for eating and shape concern, correlations between global EDE-Q score, eating concern, shape concern and weight concern vs. RWR were detected only in the high-RWR group. Shape concerns call attention to the high values in both groups, which are far from the maximum score since 2.5 is considered healthy status [28]. Severe ED symptoms can develop after surgery [35], and many cases are likely to be underreported [36]. Many patients reported food and self-image issues, as bulimic episodes, both objective, subjective and objective overeating at least once over the previous 28 days (mean 57.95% of 94 patients). The clinical relevance appears that 58.5% of patients in this sample are endorsing binge eating/loss-of-control eating by an average of 6.1 years post-surgery, although this finding is unrelated to relative weight regain. Our study showed a higher frequency of individuals who reported subjective bulimic episodes than those detected by De Zwaan et al. in patients with a mean of two years post-RYGB (approximately 25% of 59 patients) [37].

The Rep(eat)-Q showed high values of the global score on the compulsive grazing subscale and the repetitive eating subscale in the high vs. low RWR. However, positive correlations between global Rep(eat)-Q score and their subscales (compulsive and non-compulsive grazing) vs. RWR were detected for both groups. This result raises the question that the dysfunctional eating behavior of grazing may start early and remain over the years after surgery. Colles et al. reported a significant increase in grazing behavior, from 26.4% at baseline to 38.0% twelve months post-surgery [38]. Grazing is a high-risk behavior that compromises weight maintenance after bariatric surgery [18,38] and is considered a predictor of worsening weight loss trajectories [18].

Although the high-RWR group had a significantly longer time since surgery than those who regained less weight, which makes interpreting the results difficult, a significant direct association was detected only for the repetitive eating subscale in this group. Even if a causal relationship cannot be established, the extended time since surgery may have also influenced our results, rather than the eating behavior itself, exposing these patients to many weight regain causes. Our findings partially ratified this premise. Nicanor-Carreón et al. linked weight regain to sweet cravings, higher EDE-Q scores and time since surgery. [39]. These authors also detected that those with an RWR of ≥20% had increased global scores in the EDE-Q compared to those with an RWR of <20%. Some studies demonstrated an amelioration in ED symptoms after surgery, but only in the short term (1 to 3 years follow-up) [28,40,41,42]. Conversely, Conceição et al. reported an increase in the frequency of pinching or nibbling, defined as the consumption of modest amounts of food in an unplanned and repetitive manner without a sense of loss of control, from 17.5% in the first year to 47.3% two years post-surgery [43].

Some of our limitations need to be mentioned. The cross-sectional design limits our ability to establish causality. Eating habits before surgery were not investigated by us since their first visit with us was the one we collected all data for this study. However, it is known that not all those patients with preoperative disordered EBs will develop them after surgery [37]. In addition, it is known that bariatric surgery may reduce disordered EB for a while due to the anatomical alterations associated with stomach reduction [11]. Possibly influenced by changes in the gastrointestinal peptides after surgery. Furthermore, this study was based on self-reported questionnaires. Although these questionnaires can be inaccurate because of response distortions [26], standardized instruments and objective diagnostic methods for assessing ED symptoms are still limited [44]. The potential influence of other factors on recurrent weight gain, aside from anatomical causes investigated through digestive endoscopy, was not assessed in the present study. Additionally, and more importantly, two of our main limitations are the absence of pre-surgery data and the difference observed between groups in the time since surgery. The group with high RWR had a longer time since surgery, suggesting more exposition to other causes of weight regain. The loss of MHT follow-up is a novel aspect of our manuscript, and while we included the time since surgery, we presented no data on the length of time without regular healthcare follow-up. Finally, a higher proportion of females and RYGB occurred in our sample, which is commonly observed in bariatric samples. Our findings also reflect our country’s preferred choices for operative techniques [45].

## 5. Conclusions

Our study originally reported the prevalence of a range of disordered EB in a group of patients after bariatric surgery without regular healthcare follow-up, on average, six years after surgery. These patients presented high RWR and scores for EDs and grazing behavior. The high values in global EDE-Q and Rep(eat)-Q scores in both groups are noteworthy, which is of concern given the association with poorer surgical outcomes. In both low and high RWR, grazing behavior significantly correlated with RWR, but ED symptoms (except the dietary restraint) were associated with RWR only in those who regained more weight. The role of eating disorders in post-bariatric weight regain remains a significant, unresolved issue. Psychological assessment and therapy should be offered both pre- and post-surgery. Our patients reflect real-world patients. Unfortunately, many of them do not follow regular healthcare. Receiving these patients on our unit allowed us to speculate that postoperative follow-up could detect precocious disordered EB and add adherence to current recommendations, and may increase successful outcomes. Future research could shed light on our suggestion for clinical practice.

## Figures and Tables

**Table 1 nutrients-16-04250-t001:** Demographic characteristics, clinical history and bariatric surgery data of the patients.

Variable	Pooled Sample (*n* = 94)	Low RWR (*n* = 56)	High RWR (*n* = 38)	*p*-Value
*Demographic characteristics*				
Age (years)	42 ± 9	42 ± 8	42 ± 9	0.27
Female (n, %)	82 (87.2)	51 (91.1)	31 (81.6)	0.17
BMI (kg/m^2^)	32.9 ± 6.5	30.2 ± 4.2 *	36.9 ± 7.3	**<0.001**
Neck circumference (cm)	36.0 ± 3.8	35.1 ± 3.1 *	37.5 ± 4.2	**<0.001**
Waist circumference (cm)	96.6 ± 19.2	92.4 ± 12.1 *	102.9 ± 25.3	**<0.001**
Hip circumference (cm)	117.3 ± 13.9	111.8 ± 9.9 *	125.7 ± 15.1	**<0.001**
SBP (mmHg)	124.7 ± 15.7	123.4 ± 15.5	126.7 ± 16.0	0.16
DBP (mmHg)	79.2 ± 11.7	77.6 ± 12.1 *	81.6 ± 10.9	**0.05**
Heart rate (bpm)	77 ± 12	76 ± 12	79 ± 11	0.09
*Clinical history—(n, %)*				
Type 2 diabetes *mellitus*	5 (5.3)	2 (3.6)	3 (7.9)	0.36
Hypertension	15 (16.0)	8 (14.3)	7 (18.4)	0.59
Dyslipidemia	4 (4.3)	0 (0) *	4 (10.5)	**0.01**
Tobacco use	12 (12.8)	10 (17.9)	2 (5.3)	0.07
*Bariatric surgery data*				
Preoperative BMI (kg/m^2^)	48.6 ± 7.7	48.3 ± 7.0	49.0 ± 8.8	0.34
EWL (%)	88.7 ± 18.6	88.1 ± 17.3	89.6 ± 20.6	0.35
RWR (%)	22.9 ± 20.3	10.1 ± 6.7 *	41.7 ± 18.8	**<0.001**
RYGB (n, %)	80 (85.1)	50 (89.3)	30 (78.9)	0.16
Time since surgery (years)	6.1 ± 4.0	4.6 ± 2.7 *	8.3 ± 4.4	**<0.001**

SBP—Systolic blood pressure; DBP—Diastolic blood pressure; BMI—Body mass index; EWL—Excess weight loss; RWR—Ratio of weight regain; RYGB—Roux-en-Y gastric bypass; * *p*-value, unpaired Student *t*-test or chi-square test; results expressed as mean ± standard deviation and (n) %. Significant results are in bold.

**Table 2 nutrients-16-04250-t002:** Eating disorder/cognitive symptoms, and the screening of grazing of the patients.

Variable	Pooled Sample (*n* = 94)	Low RWR (*n* = 56)	High RWR (*n* = 38)	*p* Value
*EDE-Q*				
Global EDE-Q score (nbs)	2.94 ± 1.31	2.72 ± 1.37 *	3.26 ± 1.15	**0.02**
Dietary Restraint	2.33 ± 2.32	1.91 ± 1.69 *	2.53 ± 1.53	**0.01**
Eating Concern	2.17 ± 1.57	1.99 ± 1.62	2.44 ± 1.45	0.08
Shape Concern	4.18 ± 1.58	4.00 ± 1.71	4.43 ± 1.37	0.09
Weight Concern	3.26 ± 1.51	3.00 ± 1.61 *	3.66 ± 1.27	**0.02**
Objective binge eating episodes (n, %)	55 (58.5)	32 (57.1)	23 (60.5)	0.74
Subjective binge eating episodes (n, %)	55 (58.5)	32 (57.1)	23 (60.5)	0.74
Objective overeating (n, %)	54 (57.4)	32 (57.1)	22 (57.9)	0.94
Self-induced vomiting (n, %)	12 (12.8)	6 (10.7)	6 (15.8)	0.47
Excessive exercise (n, %)	15 (16)	12 (21.4)	3 (7.9)	0.08
*Rep(eat)-Q*				
Global Rep(eat)-Q score (nbs)	2.94 ± 1.67	2.62 ± 1.62 *	3.40 ± 1.65	**0.02**
Compulsive grazing subscale	2.83 ± 1.72	2.52 ± 1.73 *	3.29 ± 1.61	**0.03**
Repetitive eating subscale	2.89 ± 1.77	2.56 ± 1.74 *	3.37 ± 1.73	**0.03**

RWR—Ratio of weight regain; EDE-Q—Eating Disorder Examination Questionnaire; Rep(eat)-Q—Repetitive Eating Questionnaire; nbs—Norm-based score; n, %—Number and percentage of patients reporting the target problem at least once over the previous 28 days. * *p* value, unpaired Student *t*-test or chi-square test; results expressed as mean ± standard deviation and (n) %. Significant results are in bold.

**Table 3 nutrients-16-04250-t003:** Correlations between eating disorder symptoms vs. recurrent weight gain and time since surgery of the patients.

	Pooled Sample (*n* = 94)		Low RWR (*n* = 56)		High RWR (*n* = 38)	
	RWR	Time Since Surgery	RWR	Time Since Surgery	RWR	Time Since Surgery
*EDE-Q*						
Global EDE-Q score	**0.29** (***p* < 0.001**)	0.13 (*p* = 0.19)	0.12 (*p* = 0.36)	−0.03 (*p* = 0.79)	**0.35** (***p* = 0.03**)	0.13 (*p* = 0.42)
Dietary Restraint	0.18 (*p* = 0.06)	**0.22 ** (***p* = 0.02**)	0.01 (*p* = 0.90)	0.01 (*p* = 0.91)	0.03 (*p* = 0.83)	0.21 (*p* = 0.19)
Eating Concern	**0.26** (***p* = 0.01**)	0.08 (*p* = 0.43)	0.12 (*p* = 0.37)	−0.17 (*p* = 0.18)	**0.37** (***p* = 0.01**)	0.21 (*p* = 0.19)
Shape Concern	**0.21** (***p* = 0.03**)	0.10 (*p* = 0.32)	0.04 (*p* = 0.74)	0.002 (*p* = 0.98)	**0.31** (***p* = 0.05**)	0.09 (*p* = 0.55)
Weight Concern	**0.32** (***p* = 0.001**)	0.19 (*p* = 0.06)	0.23 (*p* = 0.07)	0.04 (*p* = 0.73)	**0.33** (***p* = 0.03**)	0.19 (*p* = 0.25)
*Rep(eat)-Q*						
Global Rep(eat)-Q score	**0.41** (***p* < 0.001**)	**0.29** (***p* < 0.001**)	**0.38** (***p* < 0.001**)	0.18 (*p* = 0.18)	**0.43** (***p* < 0.001**)	0.25 (*p* = 0.12)
Compulsive grazing subscale	**0.38** (***p* < 0.001**)	**0.22 ** (***p* = 0.03**)	**0.32 ** (***p* = 0.01**)	0.05 (*p* = 0.66)	**0.42** (***p* < 0.001**)	0.23 (*p* = 0.16)
Repetitive eating subscale	**0.32** (***p* = 0.001**)	**0.32** (***p* = 0.001**)	**0.41** (***p* = 0.001**)	0.19 (*p* = 0.15)	**0.51** (***p* < 0.001**)	**0.32 ** (***p* = 0.04**)

RWR—Ratio of weight regain; LWR—Low ratio of weight regain; HWR—High ratio of weight regain; EDE-Q—Eating Disorder Examination Questionnaire; Rep(eat)-Q—Repetitive Eating Questionnaire. Significant results are in bold.

## Data Availability

The original contributions presented in the study are included in the article, further inquiries can be directed to the corresponding author.

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
