# Peer review of "Disordered Eating Behaviors and Weight Regain in Post-Bariatric Patients"

_nutrients, 2024, doi:10.3390/nu16234250_

Round 1
Reviewer 1 Report (Previous Reviewer 1)
Comments and Suggestions for Authors
The authors have addressed my prior concenrs
Author Response
Rio de Janeiro, December 1st, 2024.
Dear Editors and Reviewers,
We would like to express our sincere gratitude for the valuable comments and suggestions provided by the reviewers. We have carefully revised the manuscript, "Disordered Eating Behaviors and Weight Regain in Patients After Bariatric Surgery" (Manuscript ID: nutrients-3283977), addressing each point raised. Furthermore, we have carefully revised the entire manuscript, incorporating all editorial suggestions to improve its scholarly merit and minimize overlap with existing literature. It should be noted that, despite numerous stylistic revisions, the semantic content of every revised sentence has been preserved.
Please refer to the tracked changes within the revised manuscript for a detailed account of the modifications made.
We trust that these revisions have addressed the reviewers’ concerns and that the manuscript is now suitable for publication.
Sincerely,
L.G. Kraemer-Aguiar On behalf of all authors
COMMENTS AND SUGGESTIONS FOR AUTHORS
Reviewer 1
The authors have addressed my prior concerns
Thank you very much for your revision. We are happy to receive your approval
Reviewer 2
The manuscript included all postulated changes. Analysing eating disorders as the cause of weight regain is an important, however well-recognised problem.
Please add a sentence that the problem remains unresolved and that psychological assessment and psychotherapy should be proposed for these patients prior as well as after bariatric surgery.
Thank you very much for your constructive revision. We added the suggested sentence in the conclusion section. The manuscript improved due to the suggestion made.
Reviewer 2 Report (Previous Reviewer 3)
Comments and Suggestions for Authors
The manuscript included all postulated changes. Analysing eating disorders as the cause of weight regain is an important, however well-recognised problem.
Please add a sentence that the problem remains unresolved and that psychological assessment and psychotherapy should be proposed for these patients prior as well as after bariatric surgery.
Author Response
Rio de Janeiro, December 1st, 2024.
Dear Editors and Reviewers,
We would like to express our sincere gratitude for the valuable comments and suggestions provided by the reviewers. We have carefully revised the manuscript, "Disordered Eating Behaviors and Weight Regain in Patients After Bariatric Surgery" (Manuscript ID: nutrients-3283977), addressing each point raised. Furthermore, we have carefully revised the entire manuscript, incorporating all editorial suggestions to improve its scholarly merit and minimize overlap with existing literature. It should be noted that, despite numerous stylistic revisions, the semantic content of every revised sentence has been preserved.
Please refer to the tracked changes within the revised manuscript for a detailed account of the modifications made.
We trust that these revisions have addressed the reviewers’ concerns and that the manuscript is now suitable for publication.
Sincerely,
L.G. Kraemer-Aguiar On behalf of all authors
COMMENTS AND SUGGESTIONS FOR AUTHORS
Reviewer 1
The authors have addressed my prior concerns
Thank you very much for your revision. We are happy to receive your approval
Reviewer 2
The manuscript included all postulated changes. Analysing eating disorders as the cause of weight regain is an important, however well-recognised problem.
Please add a sentence that the problem remains unresolved and that psychological assessment and psychotherapy should be proposed for these patients prior as well as after bariatric surgery.
Thank you very much for your constructive revision. We added the suggested sentence in the conclusion section. The manuscript improved due to the suggestion made.
This manuscript is a resubmission of an earlier submission. The following is a list of the peer review reports and author responses from that submission.
Round 1
Reviewer 1 Report
Comments and Suggestions for Authors
Overall this is a well done study that will be of interest to the field. I have one concern and one recommendation.
The concern is with multiple comparisons. The statistics section does not specify that the stats were corrected for multiple comparisons. If this was not done it needs to be. If it was done, that needs to be stated.
The recommendation is to add a paragraph summarizing the results in Table 3. Such a paragraph is provided for Table 2 and would greatly aide the reader in digesting these results.
Author Response
Rio de Janeiro, November 09th, 2024.
Dear Editors and Reviewers,
We appreciate the comments and suggestions made by reviewers. The manuscript has been revised, and itemized responses are now available. The answers are at the end of each reviewer's comment (underlined). We also thank you for the extended deadline since we were at ObesityWeek 2024 in San Antonio, TX, USA. We have performed manuscript changes and are now re-submitted with tracked changes exposed. The manuscript improved due to the suggestions made. We hope it now receives your approval for publication.
Yours Sincerely,
L.G. Kraemer-Aguiar on behalf of all authors
- Some sentences were changed without a change in meaning to avoid repetitions that could suggest plagiarism. However, we need to point out that most of the repeated sentences or words observed by the AI, from our point of view, do not denote any plagiarism since most of them express descriptive data of the authors' results or even our own methodological data. Still, we emphasize that some changes were made to the text after the request emailed by the assistant editor.
Manuscript ID: nutrients-3283977
Title: Disordered Eating behaviors and weight regain in patients after bariatric surgery
COMMENTS AND SUGGESTIONS FOR AUTHORS
Reviewer #1
(reviewer #1, comment #1): Overall this is a well done study that will be of interest to the field. I have one concern and one recommendation. The concern is with multiple comparisons. The statistics section does not specify that the stats were corrected for multiple comparisons. If this was not done it needs to be. If it was done, that needs to be stated. The recommendation is to add a paragraph summarizing the results in Table 3. Such a paragraph is provided for Table 2 and would greatly aide the reader in digesting these results.
Answer – (reviewer #1, comment #1): Thank you for your suggestions. We included more details in the statistical analysis and also added a paragraph summarizing the results of table 3
Reviewer #2
The study addresses the role of disordered eating behaviors (DEB) on weight regain following bariatric surgery. Patients lost to follow-up were re-contacted and were asked to complete a series of screening tools for DEB, together with an assessment of present anthropometry. The amount of weight regain was computed as difference from maximum weight loss (computed on the basis of recall by patients). The presence of DEB and grazing was associated with a larger amount of weight regain. I have several concerns that require adequate answer:
(reviewer #2, comment #1): As acknowledged by authors in the discussion, no data on eating behaviors before surgery were available. Thus, we cannot rule out the possibility that DEB was a chronic condition, that might reduce the adherence to post-surgery lifestyle changes. It is definitely surprising that patients were operated on without an eating behavior assessment. Authors should recollect the original data for comparison, or provide a rebuttal on the basis of the similar EWL nadir.
If DEB arose ex-novo after surgery, then the difference in post-surgery time might make the difference in RWR, reduce adherence and finally promote grazing and DEB, although they did not find a relation between RWR and DEB. Sorry to say that only sequential post-surgery eating and psychological tools might define the importance of DEB on weight trajectories. Considering that time since surgery was nearly double in weight-regainers, we might assume that this might be the final outcome of the majority of cases. Finally, we concorded with you that the time since surgery was nearly double in weight-regainers, and that this may have influenced our findings - "more time elapsed since surgery, more time without follow-up, greater weight recidivism" and we recognize this in our limitations.
Answer – (reviewer #2, comment #1): In our country, we follow the Position Statements of the Brazilian Society of Bariatric and Metabolic Surgery (Berti et al. 2015). To prepare patients who were indicated for surgical treatment of obesity, it is fundamental to have a specialized medical and multidisciplinary team. Ideally, a multidisciplinary team should comprise an endocrinologist, a surgeon highly specialized in bariatric procedures, a nutritionist doctor, a psychiatrist, a nutritionist, a psychologist, a physical trainer, a physiotherapist, and other professionals, if necessary. As we and the reviewers already know, preoperative psychological assessment allows the identification of psychopathologies that may impact surgical outcomes. Therefore, ideally, an indication for the treatment of these disorders and their stabilization before surgery would be the best practice. Other aspects should be also approached/evaluated before surgery and in our clinical daily practice at the University we always consider them preoperatively, like the patient's understanding of surgery, the reasons for undergoing surgery, the expectations regarding the postoperative results, the adherence to pre- and post-surgical recommendations, to lifestyle changes, the clinical history/evolution of weight gain and treatments performed, the presence of eating/psychiatric disorders, and the cognitive functioning. As mentioned in Methods, these patients were evaluated on their first medical visit to our Institution. Therefore, our first contact with them was done considering their clinical self-reported data, not data collected from medical charts. All of them were submitted to their surgeries outside of our Institution and lost their follow-ups, especially due to the loss of medical insurance. We do not have the original preoperative data for comparison, but it is mandatory for medical insurance (and for the public health system) to rule out psychopathologies before surgery. Therefore, we believe that in our country, all patients who have undergone surgery have a previous eating behavior assessment. Not for all, but for most, we can assure it happened. The causes related to weight regain after surgery are related, especially to behavioral ones, and should first be evaluated by a multidisciplinary team, and then the technical causes should be solved surgically (please see Berti et al. 2015). As you mentioned, one point is important to consider: the two groups had similar responses to surgery. The pooled sample had an EWL of 88.7±18.6%, but no difference existed between groups on this variable. So, surgery was effective for all, and independently of previous behavioral status, both groups got the exact nadir. Therefore, we do believe that factors related to behavioral disorders can be involved in weight recidivism (although time since surgery may have influenced the results, as already mentioned). We opted to discuss it more in the Discussion section. Another important aspect to consider in our paper is the one that shows how these patients who had undergone bariatric surgeries are following up on their treatments. We do believe that this point is of great interest to all who work with obesity.
Reference: Berti LV, Campos J, Ramos A, Rossi M, Szego T, Cohen R. Position of the SBCBM - nomenclature and definition of outcomes of bariatric and metabolic surgery. Arq Bras Cir Dig. 2015;28 Suppl 1(Suppl 1):2. doi: 10.1590/S0102-6720201500S100002. PMID: 26537262; PMCID: PMC4795295.
(reviewer #2, comment #2): I have problems with exclusion criteria, in particular with the exclusion of patients with regular physical activity at levels recommended by guidelines. The authors are invited to discuss their choice.
Answer – (reviewer #2, comment #2): Thank you for the opportunity to discuss this point. In the same line of study designs, previously, our group demonstrated that bariatric patients with higher weight regain were the ones with higher sedentary time and lower levels of physical activity (Romagna et al. 2021). Although causal relationships are not able to be done due to the cross-sectional design, several trials showed that physical inactivity contributes to weight regain following bariatric surgery. On the other hand, current position statements recommend the physical activity routines, one of the requirements of any long-term weight loss or maintenance program. Therefore, to ensure that the study design would involve, especially behavioral eating disorders and could be associated with weight recidivism, and due to the relationships with the changes in body weight and physical activity, we opted to exclude patients with regular physical activity.
We changed the text to "level of physical activity ≥ 150 min/week," which is a classification for individuals who are physically active. Although like the previous term, seemed more appropriate for a clinical manuscript. We hope it will now meet your approval.
References:
Romagna EC, Lopes KG, Mattos DMF, Farinatti P, Kraemer-Aguiar LG. Physical Activity Level, Sedentary Time, and Weight Regain After Bariatric Surgery in Patients Without Regular Medical Follow-up: a Cross-Sectional Study. Obes Surg. 2021 Apr;31(4):1705-1713. doi: 10.1007/s11695-020-05184-x. Epub 2021 Jan 6. PMID: 33409978.
El Ansari W, Elhag W. Weight Regain and Insufficient Weight Loss After Bariatric Surgery: Definitions, Prevalence, Mechanisms, Predictors, Prevention and Management Strategies, and Knowledge Gaps-a Scoping Review. Obes Surg. 2021 Apr;31(4):1755-1766. doi: 10.1007/s11695-020-05160-5.
Kushner RF, Sorensen KW. Prevention of weight regain following bariatric surgery. Curr Obes Rep. 2015;4:198–206.
Jakivic (Chair) JM, Clark K, Coleman E, et al. Appropriate intervention strategies for weight loss and prevention of weight regain for adults. Med Sci Sports Exerc. 2001;33
Executive summary: Guidelines (2013) for the management of overweight and obesity in adults: a report of the American College of Cardiology/American Heart Association Task Force on Practice Guidelines and the Obesity Society published by the Obesity Soci. Obesity (Silver Spring). 2014;22 Suppl 2:S5–39.
Mechanick JI, Youdim A, Jones DB, Timothy Garvey W, Hurley DL, Molly McMahon M, et al. Clinical practice guidelines for the perioperative nutritional, metabolic, and nonsurgical support of the bariatric surgery patient - 2013 update: Cosponsored by American Association of Clinical Endocrinologists, the Obesity Society, and American Society. Surg Obes Relat Dis. Elsevier Inc.; 2013;9:159–91.
Reviewer #3
(reviewer #3, comment #1): I would like to start the review by stating that BED is the abbreviation for binge-eating disorder and should not be used for other terms.
Answer – (reviewer #3, comment #1): Thank you for your comment. We agree with you. In our manuscript we do not use the term BED (abbreviation for binge-eating disorder), but rather DEB (abbreviation for disordered eating behaviors). We reviewed our manuscript and did not detect any incorrect use of the term. However, since you observed that we decided to change DEB to disordered EB, this change will possibly reduce the misunderstanding of the readers. Additionally, we opted to change reference 11 to another one, a more appropriate one for the theme studied in our manuscript.
Reference: Conceição EM, Goldschmidt A. Disordered eating after bariatric surgery: clinical aspects, impact on outcomes, and intervention strategies. Curr Opin Psychiatry. 2019 Nov;32(6):504-509. doi: 10.1097/YCO.0000000000000549. PMID: 31343419.
(reviewer #3, comment #2): As I understand the authors report the occurrence of eating disorders symptoms, pointing out the lack of appropriate psychological assessment before bariatric surgery. It should be clearly emphasized in the abstract.
Answer – (reviewer #3, comment #2): We thank you for the comment and included the lack of appropriate psychological assessment in the abstract. We suggest you read our answer to comment #1, reviewer #2. Possibly, it will give you more insights in respect to the design of the study.
(reviewer #3, comment #3): The discussion should be more focused on the consequences of not being diagnosed with the eating disorder before bariatric surgery.
Answer – (reviewer #3, comment #3): At this point, we also reemphasize the suggestion of reading the answer for comment #1, reviewer #2. Additionally, although eating disorders are relatively common in pre-surgery populations (1,2), there is limited evidence of the prognostic value of pre-operative eating disorders concerning post-surgical weight outcomes. Thus, a history of eating disorders may not be as effective in predicting weight outcomes. García-Ruiz-de-Gordejuela et al showed that individuals with slower weight loss trajectories in the first year after bariatric surgery presented with more pre-surgery psychopathology. On the other hand, Conceição et al showed that patients presenting with pre-operative eating disorders did not differ in weight loss trajectories across 30 months post-surgery.
References
1.Conceição EM, Orcutt M, Mitchell J, et al. Eating disorders after bariatric surgery: a case series. Int J Eat Disord. 2013;46(3):274–9.
2.Conceição EM, Utzinger LM, Pisetsky EM. Eating disorders and problematic eating behaviours before and after bariatric surgery: characterization, assessment and association with treatment outcomes. Eur Eat Disord Rev. 2015;23(6):417–25.
3.García-Ruiz-de-Gordejuela A, Agüera Z, Granero R, et al. Weight loss trajectories in bariatric surgery patients and psychopathological correlates. Eur Eat Disord Rev. 2017;25:586–94.
4.Conceição EM, Mitchell JE, Pinto-Bastos A, et al. Stability of problematic eating behaviors and weight loss trajectories after bariatric surgery: a longitudinal observational study. Surg Obes Relat Dis. 2017;13(6).
(reviewer #3, comment #4): There is also a need to describe translation to clinical practice. Is there any idea how to improve adherence after surgery?
Answer – (reviewer #3, comment #4): We believe that our results highlight the importance of adherence to regular medical follow-up in the post-surgical period, as well as the need to diagnose disordered eating behaviors in the clinical setting, especially through those easy-to-use and of low cost (as validated questionnaires for assessing the behavioral/psychosocial factors). The difficulties in the follow-up of patients with obesity are numerous, and their condition involves physiological and psychological disorders. Their emotional state can negatively influence treatment, so it is important to understand this process in the search for advances to improve care during pre-operative and post-surgical periods. Additional relevance to our findings must be considered in the scenario of an increasing number of bariatric surgeries worldwide. Such evidence is relevant for clinical practice, mainly for the medical and multidisciplinary teams. Clarifying the importance of medical consultations to better surgical results and identifying the potential barriers and facilitators of non-adherence requires the implementation of a shared decision-making process between healthcare professionals. The last 5 lines of the conclusion suggest how our study my impact healthcare for bariatric patients.
(reviewer #3, comment #5): How to prevent bariatric surgery in subjects who will not benefit from it? Certainly, the paper is of importance, however the reader should benefit from the authors' findings.
Answer – (reviewer #3, comment #5): This is precisely why we study bariatric surgery as a treatment option for obesity, as well as its outcomes and predictors. The scientific community needs to find the best predictors for better outcomes in bariatric procedures since it is one of the most common treatments employed nowadays for obesity. Additionally, it is effective and sustained for most patients who have undergone bariatric surgery. Those who regain weight are still a challenge for us, and our manuscript sheds light on this theme.
Reviewer 2 Report
Comments and Suggestions for Authors
The study addresses the role of disordered eating behaviors (DEB) on weight regain following bariatric surgery. Patients lost to follow-up were re-contacted and were asked to complete a series of screening tools for DEB, together with an assessment of present anthropometry. The amount of weight regain was computed as difference from maximum weight loss (computed on the basis of recall by patients). The presence of DEB and grazing was associated with a larger amount of weight regain. I have several concerns that require adequate answer:
1. As acknowledged by authors in the discussion, no data on eating behaviors before surgery were available. Thus, we cannot rule out the possibility that DEB was a chronic condition, that might reduce the adherence to post-surgery lifestyle changes. It is definitely surprising that patients were operated on without an eating behavior assessment. Authors should recollect the original data for comparison, or provide a rebuttal on the basis of the similar EWL nadir.
2. If DEB arose ex-novo after surgery, then the difference in post-surgery time might make the difference in RWR, reduce adherence and finally promote grazing and DEB, although they did not find a relation between RWR and DEB. Sorry to say that only sequential post-surgery eating and psychological tools might define the importance of DEB on weight trajectories. Considering that time since surgery was nearly double in weight-regainers, we might assume that this might be the final outcome of the majority of cases.
3. I have problems with exclusion criteria, in particular with the exclusion of patients with regular physical activity at levels recommended by guidelines. The authors are invited to discuss their choice.
Author Response

(The authors gave the same response as above.)

Reviewer 3 Report
Comments and Suggestions for Authors
I would like to start the review by stating that BED is the abbreviation for binge-eating disorder and should not be used for other terms.
As I understand the authors report the occurrence of eating disorders symptoms, pointing out the lack of appropriate psychological assessment before bariatric surgery. It should be clearly emphasized in the abstract.
The discussion should be more focused on the consequences of not being diagnosed with the eating disorder before bariatric surgery.
There is also a need to describe translation to clinical practice. Is there any idea how to improve adherence after surgery?
How to prevent bariatric surgery in subjects who will not benefit from it?
Certainly, the paper is of importance, however the reader should benefit from the authors' findings.
Author Response

(The authors gave the same response as above.)

Round 2
Reviewer 2 Report
Comments and Suggestions for Authors
The authors did their best to support therir conclusion, but still I believe that, in the absence of pre-surgery data, any attempt to draw conclusion is merely speculative and is biased by differences in time from surgery between groups.
We also know that bariatric surgery may reduce DEB and also BED for a while, given the difficulties associated with stomach reduction.
Reviewer 3 Report
Comments and Suggestions for Authors
The paper was improved in line with the suggestions. No further comments.